# MODEL-BASED REINFORCEMENT LEARNING WITH A HAMILTONIAN CANONICAL ODE NETWORK

## ABSTRACT

Model-based reinforcement learning usually suffers from a high sample complexity in training the world model, especially for the environments with complex dynamics. To make the training for general physical environments more efficient, we introduce Hamiltonian canonical ordinary differential equations into the learning process, which inspires a novel model of neural ordinary differential auto-encoder (NODA). NODA can model the physical world by nature and is flexible to impose Hamiltonian mechanics (e.g., the dimension of the physical equations) which can further accelerate training of the environment models. It can consequentially empower an RL agent with the robust extrapolation using a small amount of samples as well as the guarantee on the physical plausibility. Theoretically, we prove that NODA has uniform bounds for multi-step transition errors and value errors under certain conditions. Extensive experiments show that NODA can learn the environment dynamics effectively with a high sample efficiency, making it possible to facilitate reinforcement learning agents at the early stage.

## 1 INTRODUCTION

Reinforcement learning has obtained substantial progress in both theoretical foundations (Asadi et al., 2018; Jiang, 2018) and empirical applications (Mnih et al., 2013; 2015; Peters & Schaal, 2006; Johannink et al., 2019). In particular, model-free reinforcement learning (MFRL) can complete complex tasks such as Atari games (Schrittwieser et al., 2020) and robot control (Roveda et al., 2020). However, the MFRL algorithms often need a large amount of interactions with the environment (Langlois et al., 2019) in order to train an agent, which impedes their further applications. Model-based reinforcement learning (MBRL) methods can alleviate this issue by resorting to a model to characterize the environmental dynamics and conduct planning (van Hasselt et al., 2019; Moerland et al., 2020a).

In general, MBRL can quench the thirst of massive amounts of real data that may be costly to acquire, by using rollouts from the model (Langlois et al., 2019; Deisenroth & Rasmussen, 2011). It has witnessed numerous works on approximating the model with various strategies, such as least-squares temporal difference (Boyan, 1999), guided policy search (GPS) (Levine & Abbeel, 2014), dynamic Bayesian networks (DBN) (Hester & Stone, 2012), and deep neural networks (Fujimoto et al., 2018). However, the sample efficiency of MBRL can still be limited due to the high sample complexity of learning a world model when the environment is complex. Traditional methods such as the Gaussian Processes based method (Deisenroth & Rasmussen, 2011) can perform well on some problems with high sample efficiency, but they are not easy to scale to high-dimensional problems (Plaat et al., 2020). High-capacity models scale well, but they often have low sample efficiency (Plaat et al., 2020). The trade-off between scalability and sample complexity remains as a critical issue for model-based RL.

To address the aforementioned issue, we propose to introduce physical knowledge to reduce the sample complexity for learning high-dimensional dynamics in physical environments. We focus on reinforcement learning in an environment whose dynamics can be formulated by Hamiltonian canonical equations (Goldstein et al., 2002). Up till now, Hamiltonian dynamics have been successfully applied in numerous areas of physics from robotics to industrial automation. Specifically, we formulate the environments dynamics as ordinary differential equations (ODEs), and then use a

novel network architecture called Neural Ordinary Differential Auto-encoder (NODA) as our world model, which is naturally induced by physical equations.

In particular, NODA consists of two parts — an auto-encoder and an ODE network. We use the auto-encoder to get the underlying physical variables, and use the ODE network to learn the dynamics over physical variables. By using NODA, we can enjoy its ability of modeling the physical world as well as its flexibility of combining physical knowledge (if any), such as the dimension of physical variables. Theoretically, we provide uniform bounds for both multi-step transition errors and value errors for NODA by extending the former study of Lipschitz models (Asadi et al., 2018) to continuous action spaces. It is noted that NODA can be combined with both MFRL methods like SAC (Haarnoja et al., 2018) and MBRL methods like Dreamer (Hafner et al., 2019) by facilitating the learning of the world model. Extensive experiments show that we can learn NODA well using a small number of data with an appropriate structure encoded, which can boost the sample efficiency by using imaginary trajectories over the environment models (Todorov et al., 2012; Schulman et al., 2015).

## 2 BACKGROUND

We start by presenting the background knowledge of reinforcement learning, and then explain the relationship between MBRL and Hamiltonian mechanics.

### 2.1 MODEL-BASED REINFORCEMENT LEARNING

We consider the Markov decision process (MDP) model for reinforcement learning. Formally, an MDP is formulated as a tuple $\langle \mathcal{S}, \mathcal{A}, T, R, \gamma \rangle$, where $\mathcal{S}$ is the state space, $\mathcal{A}$ is the action space, $T : \mathcal{S} \times \mathcal{A} \to \mathbb{P}(\mathcal{S})$ is the transition function, $R : \mathcal{S} \times \mathcal{A} \to \mathbb{R}$ is the reward function, and $\gamma \in [0, 1)$ is a discount factor. We denote the set $\mathbb{P}(\cdot)$ as all probability measures on the space in the bracket. Our goal is to find a policy $\pi$ that can choose an action to maximize the accumulated reward. Here we focus on the challenging tasks with continuous state and action spaces (i.e., there are infinite states and actions).

MBRL aims to learn a policy by integrating planning with the aid of a known or learned model (Moerland et al., 2020b), and an essential part of MBRL is to learn a transition function that characterizes the environment. The transition function above is defined over a given state, but we can generalize it to represent the transition from a state distribution $z \in \mathbb{P}(\mathcal{S})$. By calling the generalized transition function recursively, we can get the $n$-step generalized transition function, which is defined as:

**Definition 1 (Transition Functions)** *In a metric state space $(\mathcal{S}, d_{\mathcal{S}})$ and an action space $\mathcal{A}$, we can define the generalized transition function of $T_{\mathcal{G}}$ (over state distribution $z$), and the $n$-step generalized transition function of $T_{\mathcal{G}}^n$ (for fixed sequence of actions) (Asadi et al., 2018) as*

$$
\begin{aligned}
T_{\mathcal{G}}\left(\boldsymbol{s}' \mid \boldsymbol{s}, \boldsymbol{a}\right) &= \int T\left(\boldsymbol{s}' \mid \boldsymbol{s}, \boldsymbol{a}\right) z(\boldsymbol{s}) d\boldsymbol{s} \\
T_{\mathcal{G}}^n(\cdot \mid z) &= \underbrace{T_{\mathcal{G}}\left(\cdot \mid \cdots T_{\mathcal{G}}\left(\cdot \mid z, \boldsymbol{a}_0\right) \cdots, \boldsymbol{a}_{n-1}\right)}_{n \text{ recursive calls}}.
\end{aligned}
\tag{1}
$$

Here the generalized transition gives the distribution of outcome under a certain state distribution. For MBRL, it is nontrivial to learn the transition function (i.e., the dynamics for a physical environment) because $\mathcal{S}$ can be high-dimensional. Several attempts have been made to learn such dynamics, while they have various limitations. For instance, probabilistic inference for learning control (PILCO) (Deisenroth & Rasmussen, 2011) uses Gaussian processes to model the transition function, but the inference part does not scale well to high dimensions (Langlois et al., 2019). Stochastic ensemble value expansion (STEVE) (Buckman et al., 2018) and adaptation augmented model-based policy optimization (AMPO) (Shen et al., 2020) use scalable machine learning models, but there is no further discussion on how to learn such a model efficiently. Actually, high-dimensional state and action spaces usually require much more data samples (Plaat et al., 2020). Monte Carlo tree search (MCTS) (Silver et al., 2017) can introduce human knowledge of the transition function to enhance learning, but it is restricted to cases where transition functions are totally known.

## 2.2 HAMILTONIAN MECHANICS

Methods of analytical mechanics have been introduced to predict the evolution of dynamic systems such as pendulums. For example, Lagrangian neural networks (Lutter et al., 2019; Cranmer et al., 2020) and Hamiltonian neural networks (Greydanus et al., 2019) can be used to simulate dynamic systems. These papers focus on how to model the Lagrangian or the Hamiltonian. One challenge for such methods is that the equations for the Lagrangian or the Hamiltonian are second-order differential equations, which are difficult to model in a general way. Besides, numerical solutions of second-order differential equations are prone to error accumulation.

One natural idea is to reformulate second-order differential equations into first-order ones. Then, we can use an existing neural network (Chen et al., 2018) to model these equations in the ODE form. In this paper, we concentrate on the Hamiltonian case, for which the first-order representation corresponds to Hamiltonian canonical equations (Junkins & Schaub, 2009).[1] Specifically, in Hamiltonian mechanics, we use pairs of generalized coordinate and generalized momentum $(q_k, p_k)$ to completely describe a dynamic system ($k \in \{0, 1, \cdots, K\}$), where $K$ is the dimension of generalized coordinates. It is noted that $K$ can be intuitively interpreted as the degree of freedom. We denote $p_k$ and $q_k$ as canonical states, and they are minimal independent parameters which can describe the state of the system. We define Hamiltonian $\mathcal{H} : \mathbb{R}^{2K+1} \to \mathbb{R}$ as a function of these variables and time $t$. Then the evolution of the system satisfies Hamiltonian canonical equations (Junkins & Schaub, 2009):

$$
\begin{aligned}
\frac{\mathrm{d}q_k}{\mathrm{d}t} &= \frac{\partial \mathcal{H}}{\partial p_k}, \\
\frac{\mathrm{d}p_k}{\mathrm{d}t} &= -\frac{\partial \mathcal{H}}{\partial q_k} + Q_k(t),
\end{aligned}
\tag{2}
$$

where $k \in \{0, 1, \cdots, K\}$, and $Q_k(t)$ are the generalized forces which describe the effects of external forces.

One advantage of using these equations is that they can describe general dynamic systems. Moreover, it is possible to incorporate prior knowledge into Hamiltonian canonical equations, e.g., by assuming a specific form of energy (Sprangers et al., 2014). In our case, prior knowledge can be available. For example, we may know the dimension $K$ of the generalized coordinate and generalized momentum, which only requires the knowledge of the 'degree of freedom' for a given system but makes a difference in training. Another example is about transfer learning: If we already learn the underlying dynamics, we can combine the learnt dynamics with other modules to transfer the learnt knowledge to a variety of different tasks. We empirically examine the these advantages in the experimental section.

## 3 METHODOLOGY

We now formally present NODA, which consists of an auto-encoder and an ODE network. NODA aims to serve as a simulator of the real environment (i.e., a dynamic system) by learning the transition function and the reward function. Here we assume that the transition is deterministic, otherwise an SDE network (Li et al., 2020) can be used instead. Then we can use NODA to assist reinforcement learning by generating imaginary trajectories, as outlined in Algorithm 1. Besides, we also discuss the prior knowledge that can be incorporated with NODA.

### 3.1 MODELING HAMILTONIAN CANONICAL EQUATIONS

We focus on modeling Hamiltonian canonical equations, since after getting these equations, we know the continuous evolution of the system. For many RL environments such as MuJoCo (Todorov et al., 2012), the discretization of the continuous evolution is just the transition function. However, it may be non-trivial to get these equations for a real dynamic system[2]. So we propose to use neural ODE or ODE networks (Chen et al., 2018; 2021) to model these first-order differential equations.

---

[1]Hamiltonian canonical equations are widely used with clear physical meanings and more symmetric forms than the Lagrangian case.

[2]We provide an example to show the derivation of these equations in Appendix A, where we derive these equations under the setting of Pendulum-v0 task in Gym (Brockman et al., 2016).

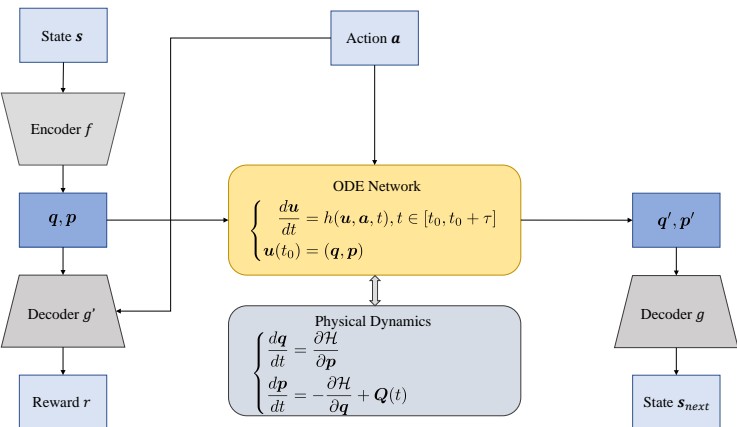

Figure 1: NODA model structure. The state $s$ goes through the encoder $f$ to get latent states $q, p$. The evolution of $u = (q, p)$ defined by the ODE network is decoded by decoder $g$ to get the next state $s_{next}$. The decoder $g'$ predicts reward $r$ using $q, p$ and action $a$.

Specifically, Neural ODE (Chen et al., 2018) provides an elegant framework to model differential equations. It is defined by $\frac{dh(t)}{dt} = f(h(t), t, \theta)$. Here $f$ is a neural network with parameter $\theta$, and we use an integrator to transform the input $h(0)$ to output $h(\tau)$, where $\tau$ is a given time horizon. Neural ODE can be viewed as a continuous version of ResNet (He et al., 2016). Here we assume that the agent affects the environment through forces. So the generalized forces $Q(t)$ in Eq. (2) correspond to the effects of action $a$ in reinforcement learning, since generalized forces can describe all physical perturbations.

## 3.2 NODA MODEL STRUCTURE

Hamiltonian canonical equations describe the evolution over canonical states, but the observed state $s$ is not necessarily composed of canonical states $q, p$. Assuming that our state $s \in \mathcal{S}$ contains the complete information of the canonical states, we firstly construct an auto-encoder to map state $s$ to canonical states $q, p$.[3] Specifically, in the auto-encoder part, we use a function $f : \mathcal{S} \to \mathbb{R}^{2K}$ to get the canonical states. Here we denote the concatenation of $(q, p)$ as $u$, so $u = f(s)$. We also need to restore the state from canonical states, so we further define a function $g : \mathbb{R}^{2K} \to \mathcal{S}$. After going through the ODE network (defined later), we can decode the evolved canonical states $s'$ to get the next state: $s_{\text{next}} = g(s')$.

Since the canonical states are concise but complete to describe the system, the auto-encoder can serve as a dimension reducer to extract and refine state $s$ when there is much redundancy in the state space. We further assume that the reward is related to the canonical states and the action. Assuming $\mathcal{A} \subset \mathbb{R}^m$, we can use another decoder $g' : \mathbb{R}^{2K} \times \mathcal{A} \to \mathbb{R}$ to get predicted reward $r$. More specifically, $r = g'(u, a)$. Then we let the predicted canonical states go through an ODE network to evolve.

In the ODE network part, the encoded latent state $u$ evolves through the ODE over a time interval $[t_0, t_0 + \tau]$. We can define the ODE network as a function $\text{ODE}(h, u, a, t_0, t_0 + \tau)$, which returns the value of the solution of neural ODE at time $t_0 + \tau$. The neural ODE is described as:

$$\frac{du}{dt} = h(u, a, t), \ t \in [t_0, \ t_0 + \tau], \tag{3}$$

where $h$ is a neural network.

After combining the ODE network and the auto-encoder, we can get a full structure of NODA, which is shown in Figure 1. The inputs of NODA are the state $s$ and the action $a$, and the outputs of NODA

---

[3]The auto-encoder is widely used in machine learning (Ballard, 1987; Ng et al., 2011; Kingma & Welling, 2013).

are the predicted state $s_{\text{next}}$ and the predicted reward $r$. Our model is formally characterized as:

$$
\begin{aligned}
s_{\text{next}} &= g(\text{ODE}(h, \boldsymbol{u}, \boldsymbol{a}, t_0, t_0 + \tau)), \\
r &= g'(\boldsymbol{u}, \boldsymbol{a}).
\end{aligned}
\tag{4}
$$

To learn the unknown parameters in $f, g$ and $h$, we define the total loss function for NODA as a convex combination of the state loss and the reward loss:

$$
\mathcal{L} = \mu(||g(\boldsymbol{u}) - \boldsymbol{s}||_2^2 + ||\boldsymbol{s}_{\text{next}} - \boldsymbol{s}_{\text{next}}^*||_2^2) + (1 - \mu)||r - r^*||_2^2,
\tag{5}
$$

where $\mu \in (0, 1)$, and $*$ denotes the ground truth. By minimizing this objective, we jointly train the RL agent and our model using interactions with the environment, and use our model together with the real environment to provide training data for the RL agent after certain training steps. More details are provided in Appendix C.

As a model of Hamiltonian canonical equations, NODA can describe general dynamic systems, and the form of canonical states can be very general. Moreover, NODA corresponds with real dynamics of the environment, which allows us to use our knowledge of the Hamiltonian. For example, we can determine $2K$ with little physical knowledge, which is the dimension of the canonical state space of a dynamic system, and we demonstrate the effects of such dimension in the experimental part. For stronger prior knowledge, we can replace the ODE network by the derived form of

---

**Algorithm 1** NODA for Reinforcement Learning

**Input:** Reinforcement learning agent $s$, environment $env$, NODA model $m$
**repeat**
    Collect data from interactions with $env$
    Use interactions with $env$ to train $m$
    Use interactions with both $env$ and $m$ to train $s$
**until** Certain number of steps
**return** Reinforcement learning agent $s$, NODA model $m$

---

Hamiltonian canonical equations, or do transfer learning between similar dynamic systems such as transferring the auto-encoder or the ODE network. We also demonstrate the effects of transfer learning in the experimental part. This makes it possible for us to make trade-offs between human expert knowledge and the required number of training samples.

## 4 THEORETICAL ANALYSIS

In this section, we present error bounds for both multi-step transition and values for NODA.

### 4.1 LIPSCHITZ DYNAMIC SYSTEMS AND NODA

Former work (Asadi et al., 2018) studied using Lipschitz function groups to model the transition of the environment. However, there are error bounds only if the real environment is Lipschitz, and that work did not mention what kind of environments have a Lipschitz transition. Theorem 1 proves that under certain conditions, the transition functions of dynamic systems are Lipschitz, which is the basis of transition and state value error bounds.

Before stating the theorem, we firstly define Lipschitz continuity, which measures the maximum magnitude of enlargement of the perturbation of input at the output side. For reinforcement learning, Definition 2 is the condition of uniformly Lipschitz continuous in the former work (Asadi et al., 2018), and it plays a central role in our theorems.

**Definition 2 (Lipschitz Models)** *In a metric state space $(\mathcal{S}, d_{\mathcal{S}})$ and an action space $\mathcal{A}$, we say that $f$ is a Lipschitz model if the Lipschitz constant (Asadi et al., 2018)*

$$
K_F := \sup_{a \in \mathcal{A}} \sup_{\boldsymbol{s}_1, \boldsymbol{s}_2 \in \mathcal{S}} \frac{d_{\mathcal{S}}(f(\boldsymbol{s}_1, \boldsymbol{a}), f(\boldsymbol{s}_2, \boldsymbol{a}))}{d_{\mathcal{S}}(\boldsymbol{s}_1, \boldsymbol{s}_2)} < \infty.
\tag{6}
$$

**Theorem 1 (Lipschitz Dynamic Systems)** *For a dynamic system with a $C^2$ continuous Hamiltonian $\mathcal{H} : \mathbb{R}^{2K+1} \to \mathbb{R}$, if the state $s$ is in a bounded closed set $\mathcal{S} \subset \mathbb{R}^l$, the evolution time equals $\tau$, the generalized force $Q_k$ is $C^1$ continuous with respect to states and bounded (for any dimension*

*k), and the transformation from states to canonical states $f^* : \mathcal{S} \to \mathbb{R}^{2K}$ is Lipschitz, then the canonical states are Lipschitz with respect to time, and the environment with respect to canonical states is Lipschitz. Additionally, if the transformation from canonical states to states $g^* : \mathbb{R}^{2K} \to \mathcal{S}$ is Lipschitz, then the environment is Lipschitz, which means (here $s \neq s'$)*

$$\sup_{a \in \mathcal{A}} \sup_{s, s' \in \mathcal{S}} \frac{d_{\mathcal{S}}\left(s_{next}, s'_{next}\right)}{d_{\mathcal{S}}\left(s, s'\right)} < \infty. \tag{7}$$

We provide a detailed proof in Appendix B, and here we only give a sketch.

*Proof.* By noting that the composition of Lipschitz functions is Lipschitz, we only need to analyse the ODE part. The conditions of $\mathcal{H}$ and $Q_k$ guarantees that the right hand side of the ODE is Lipschitz, which leads to the Lipschitz condition in this part. $\square$

Actually, the conditions in Theorem 1 are not hard to satisfy. The Hamiltonian $\mathcal{H}$ itself already has derivatives with respect to $q$, $p$ and $t$. We just further assume that the Hamiltonian $\mathcal{H}$'s derivatives with respect to $q$, $p$ and $t$ is differentiable continuous. Actually, for many dynamic systems such as spring-mass systems or three-body systems, the Hamiltonian is $C^{\infty}$ continuous.

As an imitation of Hamiltonian canonical equations, it is natural to validate if the NODA model is Lipschitz. We analyse this in Theorem 2 in which the Lipschitz model will be one of the conditions in error bounds.

**Theorem 2 (Lipschitz NODA)** *For the NODA model, if the state $s$ is in a bounded closed set $\mathcal{S} \subset \mathbb{R}^l$, $f : \mathcal{S} \to \mathbb{R}^{2K}$ is Lipschitz, the evolution time equals $\tau$, the action $a_m$ is $C^1$ continuous with respect to states and bounded (for any dimension $m$), function $h$ is $C^1$ continuous, then canonical states are Lipschitz with respect to time, and NODA with respect to canonical states is Lipschitz. Additionally, if the transformation from canonical states to states $g : \mathbb{R}^{2K} \to \mathcal{S}$ is Lipschitz, then NODA is Lipschitz, which means (here $s \neq s'$)*

$$\sup_{a \in \mathcal{A}} \sup_{s, s' \in \mathcal{S}} \frac{d_{\mathcal{S}}\left(s_{next}, s'_{next}\right)}{d_{\mathcal{S}}\left(s, s'\right)} < \infty. \tag{8}$$

The proof is in Appendix B with the similar idea as Theorem 1.

## 4.2 Uniform Error Bounds for Multi-step Transition

Here we firstly define a metric called Wasserstein Metric in Definition 3. Wasserstein metric describes how to move one distribution to another with the least cost. It has been applied to generative adversarial networks (Arjovsky et al., 2017) and reinforcement learning (Asadi et al., 2018).

**Definition 3 (Wasserstein Metric)** *In a metric space $(X, d)$, the Wasserstein metric between two probability distributions $z_1$ and $z_2$ in $\mathbb{P}(X)$ is*

$$W\left(z_1, z_2\right) := \inf_{j \in \Lambda} \iint j\left(s_1, s_2\right) d\left(s_1, s_2\right) ds_2 ds_1, \tag{9}$$

*where $\Lambda$ is a set of all joint distributions $j$ on $X \times X$ with marginal distributions $z_1$ and $z_2$ (Arjovsky et al., 2017).*

Under the conditions of Theorem 1 and Theorem 2, the dynamic system and the NODA model are Lipschitz models. On that basis, we can give uniform error bounds for multi-step transition, which is shown in Theorem 3. The theorem tells us that the multi-step transitions of the NODA model and the environment do not differ much under certain conditions, which gives support for using the NODA model as the imaginary environment.

**Theorem 3 (Transition Error Bounds)** *Under the conditions in Theorem 1 and Theorem 2, we already know that the transition function $T_{\mathcal{G}}\left(s' \mid s, a\right)$ of the environment and the transition function*

$\widehat{T}_{\mathcal{G}}\left(\boldsymbol{s}^{\prime} \mid \boldsymbol{s}, \boldsymbol{a}\right)$ *of the NODA model are Lipschitz. We denote the Lipschitz constants of these transition functions as $K_1$ and $K_2$, respectively. Let $\bar{K} = \min\{K_1, K_2\}$, then $\forall n \geq 1$:*

$$\delta(n) := W\left(\widehat{T}_{\mathcal{G}}^n(\cdot \mid \mu), T_{\mathcal{G}}^n(\cdot \mid \mu)\right) \leq \Delta \sum_{i=0}^{n-1} (\bar{K})^i, \tag{10}$$

*where $\Delta$ is an upper bound of Wasserstein metric between $\widehat{T}(\cdot \mid \boldsymbol{s}, \boldsymbol{a})$ and $T(\cdot \mid \boldsymbol{s}, \boldsymbol{a})$. This upper bound is tight for linear and deterministic transitions.*

The original theorem (Asadi et al., 2018) gives a a bound for a fixed action sequence. However, here our definitions of Lipschitz environments and models are uniform for all actions. So, by using the original proof, we give a same error bound for all possible action sequences. Thus, we get a uniform error bound under the continuous action space. This concludes the proof.

**Theorem 4** *(Value Error Bounds) Under all the conditions in Theorem 3, if the reward function $R(\boldsymbol{s})$ is uniformly Lipschitz for any action, which means we can define*

$$K_R := \sup_{a \in \mathcal{A}} \sup_{\boldsymbol{s}_1, \boldsymbol{s}_2 \in \mathcal{S}} \frac{|R(\boldsymbol{s}_1, \boldsymbol{a}) - R(\boldsymbol{s}_2, \boldsymbol{a})|}{d_{\mathcal{S}}(\boldsymbol{s}_1, \boldsymbol{s}_2)} < \infty. \tag{11}$$

*If we define state values as*

$$V_T(\boldsymbol{s}) := \sum_{n=0}^{\infty} \gamma^n \int T_{\mathcal{G}}^n(\boldsymbol{s}^{\prime} \mid \delta_{\boldsymbol{s}}) R(\boldsymbol{s}^{\prime}) d\boldsymbol{s}^{\prime}, \tag{12}$$

*where $\delta_{\boldsymbol{s}}$ means the probability that the state being $\boldsymbol{s}$ equals 1. Then $\forall \boldsymbol{s} \in \mathcal{S}$ and $\bar{K}$ (defined in Theorem 3)$\in [0, \frac{1}{\gamma}]$, we have*

$$\left|V_T(\boldsymbol{s}) - V_{\widehat{T}}(\boldsymbol{s})\right| \leq \frac{\gamma K_R \Delta}{(1-\gamma)(1-\gamma\bar{K})}. \tag{13}$$

We put the theorem of the uniform error bounds of state values here, and defer its proof and more detailed discussions of theorems to Appendix B.

## 5 EXPERIMENTS

In this section, we validate the efficiency of NODA's learning and its ability of boosting both MFRL and MBRL methods.

### 5.1 EXPERIMENTAL SETUP

**Baseline methods**  One natural baseline method for NODA is called AE, which means replacing the ODE network in NODA by an MLP. For physical simulation tasks, we choose AE and Hamiltonian neural network (HNN) (Greydanus et al., 2019) as baseline methods. For MFRL methods, we choose SAC and TD3 as our baselines. We modify the code from spinning up (Achiam, 2018) to use GPU in training. For MBRL methods, we compare our approach with state-of-the-art methods in the literature. Specifically, we focus on two methods on visual control tasks, Dreamer (Hafner et al., 2019) and BIRD (Zhu et al., 2020), as our MBRL baselines. We reproduce their results by their released codes respectively.

**Implementation**  We implement NODA mainly by using pytorch (Paszke et al., 2019) and torchdiffeq (https://github.com/rtqichen/torchdiffeq) as the implementation of the ODE network (Chen et al., 2018; 2021). We integrate NODA with MFRL methods by using it to generate imaginary trajectories and compare with MFRL baselines. When comparing with the two MBRL baselines, we combine NODA with Dreamer to get our approach: NODA-Dreamer, and we implement this in TensorFlow as Dreamer does. We use tfdiffeq (https://github.com/titu1994/tfdiffeq) as the implementation of the ODE network, which runs entirely on Tensorflow Eager Execution. We refer to Appendix C for more implementation details.

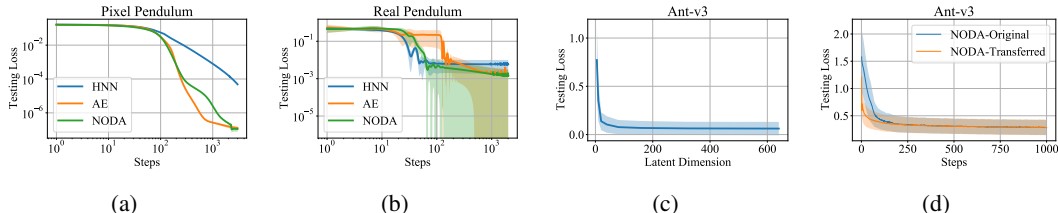

(a)           (b)           (c)           (d)

Figure 2: (a, b): Results of NODA and other physical simulators in modeling dynamic systems. (c): Sensitivity of NODA's latent dimension when learning the transition in the Ant-v3 task. (d): Transferability of NODA in the Ant-v3 task.

**Tasks** For physical simulation tasks, we use the setting of pixel pendulum and real pendulum in the paper of HNN (Greydanus et al., 2019). The pixel pendulum task aims at predicting the next frame of the pendulum using former adjacent frames, and the real pendulum task does the same thing using physical values as input. Besides, we use NODA to learn the transition part in the Ant-v3 task (actions are sampled randomly) in Gym (Brockman et al., 2016) to study the parameter sensitivity and transferability of NODA. For MFRL methods, we choose 4 MuJoCo environments in Gym to compare the performance of TD3, SAC, AE-SAC and NODA-SAC. For MBRL methods, since the two baselines themselves are evaluated on DeepMind Control Suite (DMC) (Tassa et al., 2018), we also evaluate NODA-Dreamer on 4 environments in DMC, in order to make a fair comparison with them. More experimental details can be found in Appendix D.

## 5.2 SIMULATION EFFECTIVENESS

The testing loss curves of HNN, AE and NODA over two physical environments are shown in Figure 2 (a) and (b). As the training loss functions among models can be different, we choose to compare the well-defined testing loss. In each experiment, the number of NODA's parameters equals to that of AE's parameters, and it is no more than that of HNN's parameters. In both tasks, NODA converges quickly, and achieves the best testing loss over only thousands of training iterations. It suggests that the inductive bias introduced by NODA that the system obeys Hamiltonian canonical equations accelerates training.

Figure 2 (c) shows the effect of the latent dimension on NODA's testing error after 3,000 iterations. Here we use NODA to learn the transition part in the Ant-v3 task. By mechanics, we know that the dimension of Hamiltonian canonical equations (corresponding to the dimension of the latent space in NODA) is approximately 40 in this task. We can see that a latent dimension of 40 is sufficient for NODA to achieve a high performance. This suggests that if we know the approximate dimension, we can use the prior knowledge to get a sweet point.

Figure 2 (d) shows the testing loss of the original NODA and the transferred NODA in the Ant-v3 task. We train the latter model for 100 steps over the one step transition of Ant-v3, and the final task is to predict the two-step transition. The figure suggests that we can transfer NODA over similar environments to accelerate training.

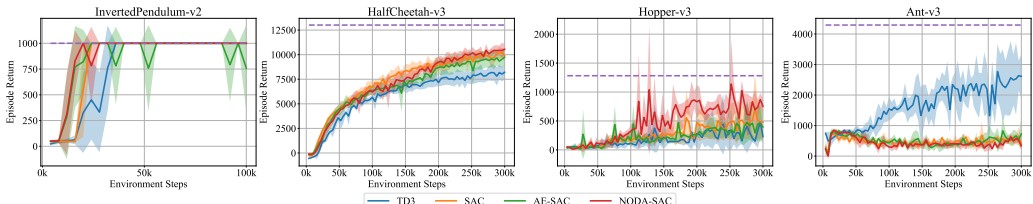

Figure 3: Results of TD3, SAC, AE-SAC and NODA-SAC on MuJoCo environments in Gym. The models are evaluated every 4k steps for 10 episodes, and the means and standard deviations are computed across 10 episodes across 4 seeds. The dashed lines are the best asymptotic performances of 1M steps.

Table 1: Experiments in MuJoCo environments (200k steps, 4 seeds)

| Algorithms | Environments | | | |
|---|---|---|---|---|
| MFRL Algorithms | InvertedPendulum | HalfCheetah | Hopper | Ant |
| TD3 | **1000.0±0.0** | 7154.8±364.9 | 283.1±245.8 | **2218.8±539.4** |
| SAC | **1000.0±0.0** | 9150.5±642.9 | 476.9±297.6 | 492.3±209.4 |
| AE-SAC | **1000.0±0.0** | 8703.0±738.4 | 255.6±34.4 | 400.5±112.2 |
| NODA-SAC | **1000.0±0.0** | **9241.8±567.7** | **832.5±283.4** | 402.4±54.9 |
| MBRL Algorithms | Hopper-Stand | Walker-Walk | Finger-Spin | Walker-Run |
| Dreamer | 198.0±254.3 | 642.9±58.5 | 366.2±252.5 | **229.1±11.4** |
| BIRD | 236.4±138.2 | 601.6±37.9 | 361.7±246.8 | 216.0 ±23.4 |
| NODA-Dreamer | **260.2±284.9** | **764.9±146.9** | **428.5±96.3** | 224.2±34.7 |

## 5.3 SAMPLE EFFICIENCY

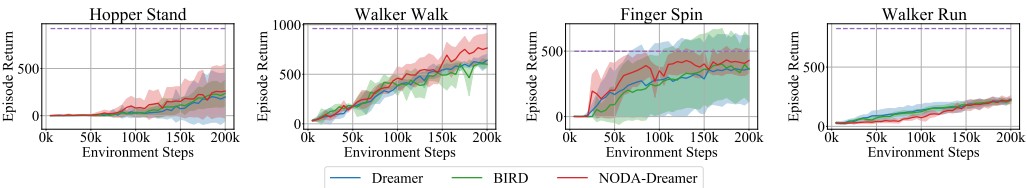

Figure 4: Results of Dreamer, BIRD and NODA-Dreamer on DMC. The models are evaluated every 5k steps for 10 episodes, and the means and standard deviations are computed across 10 episodes across 4 seeds. The dashed lines are the asymptotic performances of 5M steps of Dreamer mentioned in (Hafner et al., 2019).

The episode returns achieved by MFRL methods and MBRL methds (with/without NODA) are shown in Table 1, Figure 3 and Figure 4. For MFRL methods, NODA-SAC achieves the best performance with good robustness (see the InvertedPendulum-v2 task in Figure 3). This means that NODA can enhance the performance of the state-of-the-art algorithm SAC by imaginary trajectories. Meanwhile, if we substitute the ODE part using an MLP (denoted as the AE-SAC model), the performance will fall. This suggests that introducing an ODE network, i.e., the prior knowledge of the existence of Hamiltonian indeed helps. But for environments that SAC do not perform well (such as the Ant-v3 task), the enhancement of NODA is not significant. For MBRL methods, NODA-Dreamer can also achieve a higher return than BIRD and Dreamer in many environments. In the task 'Walker Run', during the first half of training, NODA-Dreamer lags far behind Dreamer, but it quickly catches up in the second half. This suggests that NODA has the power to accelerate training after some warm-up steps. Generally, NODA can enhance the performance for both MFRL and MBRL methods. More analyses and experiments can be found in Appendix D.

## 6 CONCLUSION

This paper proposes a novel simulator called NODA for reinforcement learning. Motivated by Hamiltonian canonical equations in physics, NODA has clear physical meanings. This allows us to incorporate prior knowledge or do transfer learning, which can improve sample efficiency. In the theoretical part, we fill the gap between former theorems about Lipschitz models and the physical world by proving that dynamic systems can be Lipschitz and extending former limited theorems to continuous action spaces. Besides, we give uniform transition error bounds and value error bounds for NODA. In the experimental part, we verify that NODA provides a more efficient modeling than HNN, and we can use prior knowledge or transfer learning to further boost its training. Not only is NODA itself sample-efficient, but NODA can improve the sample efficiency of both MFRL and MBRL methods such as SAC and Dreamer.

## 7 Ethics Statement

In this paper, we only use data sets that we are permitted to use, and we present our method and results in a transparent, honest and reproducible way. We fully acknowledge any contributions to this work. The possible harm that our method can bring originates from the modeling errors of NODA, which can lead the agent to go wrong. However, this can be mitigated by theoretical guarantees as well as human supervision. Generally, we believe that our work will make reinforcement learning more accessible by reducing the number of training samples that may be costly to acquire.

## 8 Reproducibility Statement

We give formal proofs for our theorems in Appendix B. Our code is provided in supplementary material. The NODA folder contains experiments about simulation effectiveness and experiments in Gym environments, and the NODA-Dreamer folder contains experiments in DMC environments.

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

## A   HAMILTONIAN CANONICAL EQUATIONS: PENDULUM-V0 AS AN EXAMPLE

The settings of the Pendulum-v0 task can be found on GitHub[4].

Let $x$ be the axis horizontally to the right, $y$ be the axis stright up. The pendulum's position can be described by generalized position $\theta$, where $\theta$ is the angle from the $y$ axis to the pendulum.

For the free end, $x = -l\sin\theta$, $y = l\cos\theta$. For a force perpendicular to the pendulum (the angle from the $y$ axis to $u$ is $\theta + \pi/2$) with its 2-norm equaling $u$, its generalized force $Q = \frac{\partial r}{\partial x}u_x + \frac{\partial r}{\partial y}u_y = -l\cos\theta \cdot (-u\cos\theta) - l\sin\theta \cdot (-u\sin\theta) = ul$, which is just the torque of the force.

---

[4]`https://github.com/openai/gym/blob/master/gym/envs/classic_control/pendulum.py`

The kinetic energy $T = 1/6ml^2(\mathrm{d}\theta/\mathrm{d}t)^2$, and the potential energy $V = 1/2mgl\cos\theta$, so we get the Lagrangian in Equation (14).

$$\mathcal{L}\left(\theta, \frac{\mathrm{d}\theta}{\mathrm{d}t}, t\right) = T - V = \frac{1}{6}ml^2\left(\frac{\mathrm{d}\theta}{\mathrm{d}t}\right)^2 - \frac{1}{2}mgl\cos\theta \tag{14}$$

We denote $\theta$ as $q$, the canonical momentum for $\theta$ as $p$, then

$$p = \frac{\partial\mathcal{L}}{\partial\left(\frac{\mathrm{d}q}{\mathrm{d}t}\right)} = \frac{1}{3}ml^2\frac{\mathrm{d}q}{\mathrm{d}t} \tag{15}$$

Then we can write the Hamiltonian in Equation (16).

$$\mathcal{H}(q, p, t) = \frac{\mathrm{d}q}{\mathrm{d}t}p - \mathcal{L} = \frac{3p^2}{2ml^2} + \frac{1}{2}mgl\cos q \tag{16}$$

Then Hamiltonian canonical equations give out the dynamics, as is shown in Equation (17).

$$\begin{cases} \dfrac{\mathrm{d}q}{\mathrm{d}t} = \dfrac{\partial\mathcal{H}}{\partial p} = \dfrac{3p}{ml^2} \\ \dfrac{\mathrm{d}p}{\mathrm{d}t} = -\dfrac{\partial\mathcal{H}}{\partial q} + Q(t) = \dfrac{1}{2}mgl\sin q + u(t) \end{cases} \tag{17}$$

This is the general case, but the code implementation is a little different. In the code, $u(t)$ is clipped to $[-2, 2]$, $\theta$ is clipped to $[-\pi, \pi]$, and $\mathrm{d}\theta/\mathrm{d}t$ is clipped to $[-8, 8]$. This exactly gives the bounds of the state space, since states are $[\cos\theta, \sin\theta, \mathrm{d}\theta/\mathrm{d}t]$. The clips can be added to Hamiltonian canonical equations, so the code implementation can be viewed as a solver for Equation (18).

$$\begin{cases} \dfrac{\mathrm{d}q}{\mathrm{d}t} = \dfrac{3p}{ml^2} \\ \dfrac{\mathrm{d}p}{\mathrm{d}t} = \begin{cases} \min(\dfrac{1}{2}mgl\sin q + \mathrm{clip}(u(t), -2, 2), 0), \text{ if } p > \dfrac{8}{3ml^2} \\ \max(\dfrac{1}{2}mgl\sin q + \mathrm{clip}(u(t), -2, 2), 0), \text{ if } p < -\dfrac{8}{3ml^2} \\ \dfrac{1}{2}mgl\sin q + \mathrm{clip}(u(t), -2, 2), \text{ otherwise} \end{cases} \end{cases} \tag{18}$$

The encoder $f$ is a function from $[\cos\theta, \sin\theta, \mathrm{d}\theta/\mathrm{d}t]$ to $[\theta, 1/3ml^2 \cdot \mathrm{d}\theta/\mathrm{d}t]$, which is invertible and Lipschitz. Similarly, the decoder $g$ is Lipschitz. Besides, the right hand side of Hamiltonian canonical equations is continuous and bounded.

## B  PROOFS

**Theorem 5** *(Lipschitz Dynamic Systems) For a dynamic system with a $C^2$ continuous Hamiltonian $\mathcal{H} : \mathbb{R}^{2K+1} \to \mathbb{R}$, if the state $s$ is in a bounded closed set $\mathcal{S} \subset \mathbb{R}^l$, the evolution time equals $\tau$, the generalized force $Q_k$ is $C^1$ continuous with respect to states and bounded (for any dimension $k$), and the transformation from states to canonical states $f^* : \mathcal{S} \to \mathbb{R}^{2K}$ is Lipschitz, then the canonical states are Lipschitz with respect to time, and the environment with respect to canonical states is Lipschitz. Additionally, if the transformation from canonical states to states $g^* : \mathbb{R}^{2K} \to \mathcal{S}$ is Lipschitz, then the environment is Lipschitz, which means (here $s \neq s'$)*

$$\sup_{a\in\mathcal{A}} \sup_{s,s'\in\mathcal{S}} \frac{d_\mathcal{S}\left(s_{next}, s'_{next}\right)}{d_\mathcal{S}\left(s, s'\right)} < \infty. \tag{19}$$

*Proof.* We know that $u = f^*(s)$, where $u$ is the concatenation of $q$ and $p$. Because $s$ is in a bounded closed set, and function $f^*$ is continuous (a Lipschitz function is continuous), each dimension of $f^*$ has a maximum value and a minimum value. So each dimension of $f^*$ is bounded, which means each dimension of $q$ and $p$ is bounded.

Now we are going to look into Hamiltonian canonical equations. Because $\mathcal{H}$ is $C^2$ continuous, $\frac{\partial\mathcal{H}}{\partial p_k}$ and $-\frac{\partial\mathcal{H}}{\partial q_k}$ are $C^1$ continuous (for each $k \in \{1, \cdots, K\}$). Because $q_k$ and $p_k$ are bounded for all

the values of $k$, the concatenation of $\boldsymbol{p}$, $\boldsymbol{q}$ and $t$ lies in a bounded closed set. So $\frac{\partial \mathcal{H}}{\partial p_k}$ and $-\frac{\partial \mathcal{H}}{\partial q_k}$ have a maximum value and a minimum value, which means they are bounded. Note that $Q_k$ is also bounded, then the right hand side of Hamiltonian canonical equations is bounded.

We denote the bound of $\left|\frac{\partial \mathcal{H}}{\partial p_k}\right|$ as $M_k$, then we can give the Lipschitz constant for $q_k$:

$$|q_k(t_2) - q_k(t_1)| = \left|\frac{\partial \mathcal{H}}{\partial p_k}\bigg|_{t=\xi}(t_2 - t_1)\right| \leq M_k|t_2 - t_1|. \tag{20}$$

Here $\xi$ is a value between $t_1$ and $t_2$.

Similarly, we denote the bound of $\left|-\frac{\partial \mathcal{H}}{\partial p_k} + Q_k\right|$ as $M_k'$, then we can give the Lipschitz constant for $p_k$:

$$|p_k(t_2) - p_k(t_1)| = \left|\left(-\frac{\partial \mathcal{H}}{\partial p_k} + Q_k\right)\bigg|_{t=\xi}(t_2 - t_1)\right| \leq M_k'|t_2 - t_1|. \tag{21}$$

From Equation (20) and Equation (21), we know that the canonical states are Lipschitz with respect to time. Let the right hand side of Hamiltonian canonical equations be $\boldsymbol{I}(\boldsymbol{u}, t)$. Since $\mathrm{d}\boldsymbol{I}(\boldsymbol{u}, t)/\mathrm{d}\boldsymbol{u}$ is continuous, and $\boldsymbol{u}$ and $t$ lie in a bounded closed set, $\mathrm{d}\boldsymbol{I}(\boldsymbol{u}, t)/\mathrm{d}\boldsymbol{u}$ is bounded. Then for different initial canonical states $\boldsymbol{u}_{t_0}$ and $\boldsymbol{u}'_{t'_0}$, we have (here the absolute value and the supremum for a vector/matrix is taken over each element of a vector/matrix, and the $\leq$ between two vectors means the relation between each element):

$$\left|\frac{\mathrm{d}(\boldsymbol{u}_{t_0}(t) - \boldsymbol{u}'_{t'_0}(t))}{\mathrm{d}t}\right| = |\boldsymbol{I}(u_{t_0}(t), t) - \boldsymbol{I}(u'_{t'_0}(t), t)|$$

$$\leq \sup_{\boldsymbol{u},t}\left|\frac{\mathrm{d}\boldsymbol{I}(\boldsymbol{u}, t)}{\mathrm{d}\boldsymbol{u}}\right| |u_{t_0}(t) - u'_{t'_0}(t)| \tag{22}$$

$$\leq \left\|\sup_{\boldsymbol{u},t}\left|\frac{\mathrm{d}\boldsymbol{I}(\boldsymbol{u}, t)}{\mathrm{d}\boldsymbol{u}}\right|\right\|_\infty |u_{t_0}(t) - u'_{t'_0}(t)|.$$

So

$$d_\infty(\boldsymbol{u}_{t_0}(t + \tau), \boldsymbol{u}'_{t'_0}(t + \tau)) \leq \exp\left\{\tau \left\|\sup_{\boldsymbol{u},t}\left|\frac{\mathrm{d}\boldsymbol{I}(\boldsymbol{u}, t)}{\mathrm{d}\boldsymbol{u}}\right|\right\|_\infty\right\} \cdot d_\infty(\boldsymbol{u}_{t_0}, \boldsymbol{u}'_{t'_0}). \tag{23}$$

Since $\frac{\mathrm{d}\boldsymbol{I}(\boldsymbol{u},t)}{\mathrm{d}\boldsymbol{u}}$ is bounded, $\exp\left\{\tau \left\|\sup_{\boldsymbol{u},t}\left|\frac{\mathrm{d}\boldsymbol{I}(\boldsymbol{u},t)}{\mathrm{d}\boldsymbol{u}}\right|\right\|_\infty\right\} < \infty$. This means the canonical states are Lipschitz with respect to initial canonical states. Because the composition of two Lipschitz functions is Lipschitz and $f^*$ is Lipschitz, we know that the environment is Lipschitz with respect to canonical states.

If we further assume that $g^*$ is Lipschitz, we know the environment is Lipschitz uniformly for all the action $\boldsymbol{a}$ with respect to states (the composition of two Lipschitz functions is Lipschitz), which is just Equation (19). This concludes the proof. $\square$

**Theorem 6** *(Lipschitz NODA) For the NODA model, if the state $\boldsymbol{s}$ is in a bounded closed set $\mathcal{S} \subset \mathbb{R}^l$, $f : \mathcal{S} \to \mathbb{R}^{2K}$ is Lipschitz, the evolution time equals $\tau$, the action $a_m$ is $C^1$ continuous with respect to states and bounded (for any dimension $m$), function $h$ is $C^1$ continuous, then canonical states are Lipschitz with respect to time, and NODA with respect to canonical states is Lipschitz. Additionally, if the transformation from canonical states to states $g : \mathbb{R}^{2K} \to \mathcal{S}$ is Lipschitz, then NODA is Lipschitz, which means (here $\boldsymbol{s} \neq \boldsymbol{s}'$)*

$$\sup_{\boldsymbol{a} \in \mathcal{A}} \sup_{\boldsymbol{s}, \boldsymbol{s}' \in \mathcal{S}} \frac{d_\mathcal{S}(\boldsymbol{s}_{next}, \boldsymbol{s}'_{next})}{d_\mathcal{S}(\boldsymbol{s}, \boldsymbol{s}')} < \infty. \tag{24}$$

*Proof.* The proof is similar to the proof of Theorem 5, and here we just give a sketch. We know $\boldsymbol{u} = f(\boldsymbol{s})$, where $\boldsymbol{s}$ is in a bounded closed set. Because function $f$ is continuous, each dimension of $\boldsymbol{q}$ and $\boldsymbol{p}$ is bounded.

Because $q_k$ and $p_k$ are bounded for all the values of $k$, the concatenation of $\boldsymbol{q}$, $\boldsymbol{p}$, $t$ and $\boldsymbol{a}$ lies in a bounded closed set. As a result, the output of the continuous function $h$ is bounded. We denote the bound of the $k$th output of the right hand side of the ODE as $W_k$.

$$|u_k(t_2) - u_k(t_1)| \leq W_k|t_2 - t_1| \tag{25}$$

Equation (25) tells us that $\boldsymbol{u}$ is Lipschitz with respect to time $t$.

After that, we can just follow the corresponding part (the part after Equation (21)) in Theorem 5's proof to get final results. This concludes the proof. $\square$

With this bound of multi-step transition, we can give bounds for state values. These bounds tell us that under certain conditions, the optimal policy learned from the simulator and the optimal policy learned from the environment do not differ much.

**Theorem 7** *(Transition Error Bounds) Under the conditions in Theorem 5 and Theorem 6, we already know that the transition function $T_{\mathcal{G}}\left(\boldsymbol{s}' \mid \boldsymbol{s}, \boldsymbol{a}\right)$ of the environment and the transition function $\widehat{T}_{\mathcal{G}}\left(\boldsymbol{s}' \mid \boldsymbol{s}, \boldsymbol{a}\right)$ of the NODA model is Lipschitz. We denote the Lipschitz constant of these transition functions as $K_1$ and $K_2$. Let $\bar{K} = min\{K_1, K_2\}$, then $\forall n \geq 1$:*

$$\delta(n) := W\left(\widehat{T}_{\mathcal{G}}^n(\cdot \mid \mu), T_{\mathcal{G}}^n(\cdot \mid \mu)\right) \leq \Delta \sum_{i=0}^{n-1}(\bar{K})^i. \tag{26}$$

*Here $\Delta$ is defined as the upper bound of Wasserstein metric between $\widehat{T}\left(\cdot \mid \boldsymbol{s}, \boldsymbol{a}\right)$ and $T\left(\cdot \mid \boldsymbol{s}, \boldsymbol{a}\right)$.*

The original theorem (Asadi et al., 2018) gives a a bound for a fixed action sequence. However, here our definitions of Lipschitz environments and models are uniform for all actions. So, by using the original proof, we give a same error bound for all possible action sequences. Thus, we get a uniform error bound under the continuous action space. This concludes the proof. Now we are going to give bounds for state values.

**Theorem 8** *(Value Error Bounds) Under all the conditions in Theorem 7, if the reward function $R(\boldsymbol{s})$ is (uniformly) Lipschitz, which means we can define*

$$K_R := \sup_{a \in \mathcal{A}} \sup_{\boldsymbol{s}_1, \boldsymbol{s}_2 \in \mathcal{S}} \frac{|R\left(\boldsymbol{s}_1, \boldsymbol{a}\right) - R\left(\boldsymbol{s}_2, \boldsymbol{a}\right)|}{d_{\mathcal{S}}\left(\boldsymbol{s}_1, \boldsymbol{s}_2\right)} < \infty. \tag{27}$$

*If we define state values as*

$$V_T(\boldsymbol{s}) := \sum_{n=0}^{\infty} \gamma^n \int T_{\mathcal{G}}^n\left(\boldsymbol{s}' \mid \delta_{\boldsymbol{s}}\right) R\left(\boldsymbol{s}'\right) d\boldsymbol{s}', \tag{28}$$

*where $\delta_{\boldsymbol{s}}$ means the probability that the state is $\boldsymbol{s}$ equals 1. Then $\forall \boldsymbol{s} \in \mathcal{S}$ and $\bar{K}$ (defined in Theorem 7)$\in [0, \frac{1}{\gamma}]$, we have*

$$\left|V_T(\boldsymbol{s}) - V_{\widehat{T}}(\boldsymbol{s})\right| \leq \frac{\gamma K_R \Delta}{(1 - \gamma)(1 - \gamma\bar{K})}. \tag{29}$$

The original theorem (Asadi et al., 2018) is for action space $\mathcal{A} = \{\boldsymbol{a}\}$, which means there is only one action. Besides, the original theorem assumes that the reward function only depends on state $\boldsymbol{s}$.

However, these strict limitations can be overcome by our uniformly Lipschitz models and requiring a uniformly Lipschitz reward function, as is shown in Equation (27). As long as these conditions are satisfied, we can just follow the path of the original proof. Specifically, for any action sequence, Equation (29) holds, so we get a uniform error bound. This concludes the proof.

Here we can find that a crucial condition in the proofs of Theorem 7 and Theorem 8 is that both the model and the environment is Lipschitz. In fact, Theorem 5 and Theorem 6 lay the foundations of uniform transition error bounds and value error bounds.

## C    ALGORITHM DETAILS

Algorithm 2, 3 and 4 show how we combine SAC (Haarnoja et al., 2018) with NODA. Generally, we use interactions with the environment to train a NODA model, and use it to generate imaginary trajectories to facilitate the training of SAC. We combine AE with SAC in the same way. For efficiency, we run 50 interactions with the real environment, and then update the agent and our model 50 times, which is also implemented by spinning up (Achiam, 2018). We also reduce the number of imaginary training batches per epoch when interactions with the real environment is sufficient.

Combining NODA with MBRL methods is easier. For Dreamer (Hafner et al., 2019), it itself works in an auto-encoder's fashion, that is, it encodes observations and actions into compact latent states and makes decisions and predictions in the latent state space using its world model, after which the observations are reconstructed via representation learning. Therefore, in order to combine NODA with Dreamer, we just need to replace the deterministic path(a GRU cell) of the recurrent state space model(RSSM) in Dreamer with an ODE network, and we get NODA-Dreamer. The input compact state to the original deterministic state model in Dreamer, which is believed to be a canonical state, evolves by going through the ODE network. We also similarly implement NODA-BIRD by replacing the deterministic path of the RSSM in BIRD (Zhu et al., 2020) with an ODE network, since BIRD and Dreamer have the same RSSM. All other algorithm details remain the same as in Dreamer and in BIRD respectively.

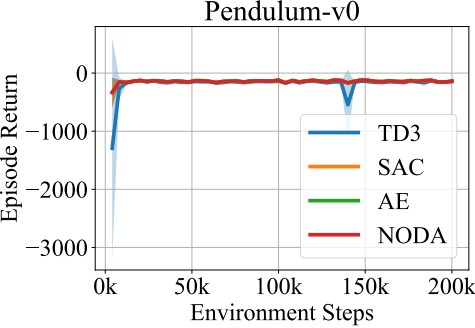

Figure 5: Results of TD3, SAC, AE and NODA on the Pendulum-v0 environment in Gym. The models are evaluated every 4k steps for 10 episodes, and the means and standard deviations are computed across 10 episodes across 4 seeds. The performance of NODA-SAC is stable, and NODA is able to facilitate the training of SAC in an early stage.

## D    EXPERIMENTAL DETAILS

In the pixel pendulum task, we run all the models for 3,000 batches with a batch size of 200. we decode the output of $h$ as the velocity prediction and set the next state to the evolution of the current state after a very short time (assuming the velocity does not change) as a weak supervision. More details can be found in our code. In the real pendulum task, we run all the models for 2,000 batches with a batch size of 200. The learning rate for both tasks is 1e-3.

When comparing different NODA models in the Ant-v3 task, we generate 20,000 steps for training and 20,000 steps for testing by a random policy. Each NODA model is trained by 3,000 batches. The batch size is 256, and the learning rate is still 1e-3.

For TD3 (Fujimoto et al., 2018), SAC (Haarnoja et al., 2018), AE-SAC and NODA-SAC, we use the code provided by spinning up (Achiam, 2018) (the PyTorch (Paszke et al., 2019) version). We modify it to use GPU, and use batch size=100 for TD3 and SAC (original setting), and batch size=256 for AE-SAC and NODA-SAC. Other parameters are the same as what are used in spinning up. We

---

**Algorithm 2** NODA-SAC for Reinforcement Learning

---

**Input:** SAC actor $\pi$, SAC critics $q_1$, $q_2$, environment $env$, NODA model $m$ (with optimizer $opt_m$), warmup steps $N_1$, model generate intervals $N_2$, model planning steps $N_3$, interaction buffer $N_4$, batch size $B_1$, model batch size $B_2$, SAC training function $T_1$, NODA training function $T_2$, data generation function for NODA $G$

$done \leftarrow True$
$step \leftarrow 0$
$D \leftarrow [\,]$
$D_m \leftarrow [\,]$
**repeat**
  **if** $done$ **then**
    $s \leftarrow env.reset()$
  **end if**
  **if** $step \leq N_1$ **then**
    $a \leftarrow env.sample(B_1)$
    $s_2, r, done \leftarrow env.step(a)$
    $D.append(\{s, a, s_2, r, done\})$
  **else**
    $a \leftarrow \pi(s)$
    $s_2, r, done \leftarrow env.step(a)$
    $D.append(\{s, a, s_2, r, done\})$
    $batch \leftarrow D.sample(B_1)$
    $\pi, q_1, q_2 \leftarrow T_1(\pi, q_1, q_2, batch)$
    $m \leftarrow T_2(m, batch, opt_m)$
    **if** $step \% N_2 == 0$ **then**
      $D_m \leftarrow G(\pi, q_1, q_2, m, D, D_m, B_2, N_3)$
      $batch \leftarrow D.sample(B_1)$
      $\pi, q_1, q_2 \leftarrow T_1(\pi, q_1, q_2, batch)$
    **end if**
  **end if**
  $step \leftarrow step + 1$
**until** $step = N_4$
**return** SAC actor $\pi$, SAC critics $q_1$, $q_2$, NODA model $m$

---

**Algorithm 3** Interaction function $G$ in NODA-SAC

---

**Input:** SAC actor $\pi$, SAC critics $q_1$, $q_2$, NODA model $m$, environment buffer $D$, model buffer $D_m$, batch size $B$, model planning steps $N$, learning rate $lr$ (default=1e-5)

$batch \leftarrow D.sample(B)$
$s, a \leftarrow batch.s, batch.a$
$planning\_step \leftarrow 0$
**repeat**
  $s_2, r, done \leftarrow m.step(s, a)$
  $D_m.append(\{s, a, s_2, r, done\})$
  $s \leftarrow s_2$
  $a \leftarrow \pi(s)$
  $planning\_step \leftarrow planning\_step + 1$
**until** $planning\_step = N$
**return** model buffer $D_m$

---

implement NODA by PyTorch, and we use torchdiffeq[5] as the implementation of the ODE network (Chen et al., 2018; 2021).

In addition to MuJoCo (Todorov et al., 2012; Schulman et al., 2015) environments, we also compare these algorithms over a simple physical environment, Pendulum-v0 in Gym (Brockman et al., 2016),

---

[5]https://github.com/rtqichen/torchdiffeq

---

**Algorithm 4** Training function $T_2$ for NODA

---

**Input:** NODA model $m$, a batch of buffer $batch$, optimizer $opt$
$s, a, s_2, r \leftarrow batch.s, batch.a, batch.s_2, batch.r$
$loss_{srecon} \leftarrow (||m.decoder(m.encoder(s)) - s||_2^2).mean()$
$m.set\_state(s)$
$s_2', r' \leftarrow m.step(a)$
$loss_{spred} \leftarrow (||s_2' - s||_2^2).mean()$
$loss_s \leftarrow loss_{srecon} + loss_{spred}$
$loss_r \leftarrow (||r' - r||_2^2).mean()$
$loss \leftarrow 0.5 \cdot loss_s + 0.5 \cdot loss_r$
$opt.zero\_grad()$
$loss.backward()$
$opt.step()$
**return** NODA model $m$

---

and the result is shown in Figure 5. It shows that the return of NODA-SAC converges quickly, and is quite stable.

For NODA-Dreamer, all the experimental details remain the same as those in Dreamer, except that we run only 200,000 steps for every task we try, and we evaluate the performance of the model every 5,000 steps. What's more, we use tfdiffeq[6] as the implementation of the Neural ODE, which runs entirely on Tensorflow Eager Execution. We evaluate Dreamer, BIRD and NODA-Dreamer on 6 tasks in total in Figure 6. We also evaluate NODA-BIRD, which is mentioned in the previous section, on 3 tasks, as shown in Figure 7.

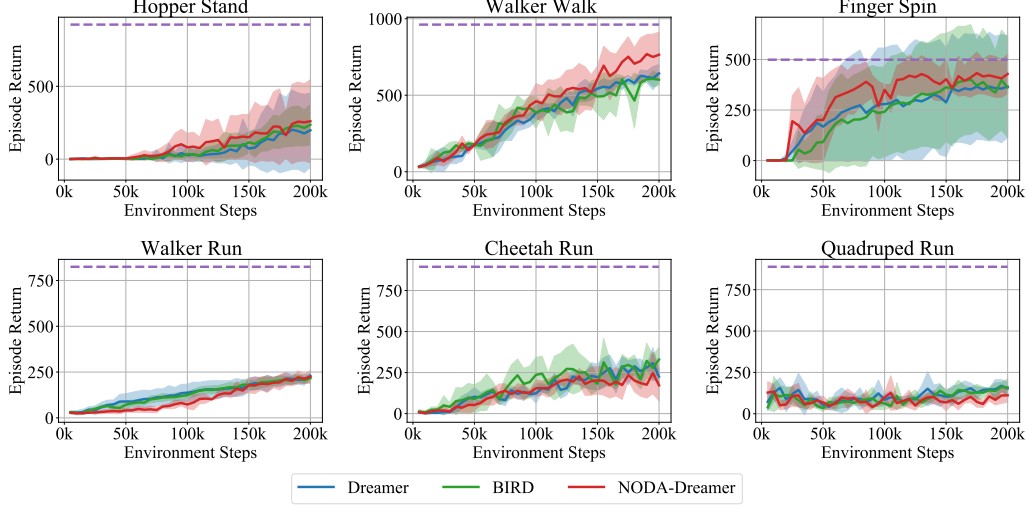

Figure 6: Results of Dreamer, BIRD and NODA-Dreamer on 6 tasks in DeepMind Control Suite. The models are evaluated every 5k steps for 10 episodes, and the means and standard deviations are computed across 10 episodes across 4 seeds. The dashed lines are the asymptotic performances of 5M steps of Dreamer mentioned in (Hafner et al., 2019). In many tasks, NODA-Dreamer has higher episode return starting early in the training process, which shows that NODA can assist in improving sample efficiency. In "Walker Run", during the first half of training, NODA-Dreamer lags far behind Dreamer and BIRD, but it quickly catches up in the second half. This suggests that NODA has the power to accelerate training after some warm-up steps. In "Quadruped Run", 200,000 steps are not sufficient for any algorithm to learn meaningful information.

---

[6]https://github.com/titu1994/tfdiffeq

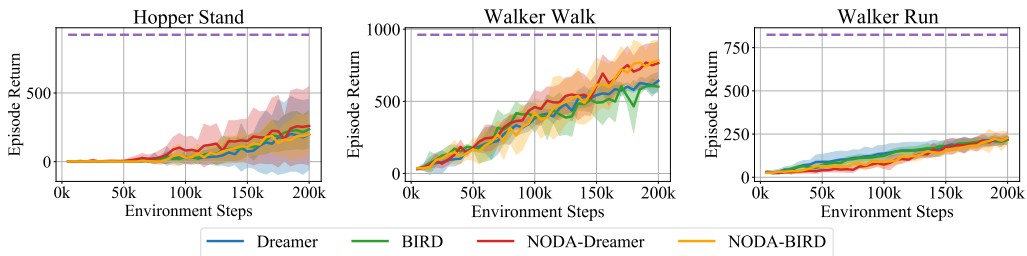

Figure 7: Results of Dreamer, BIRD, NODA-Dreamer and NODA-BIRD on 3 tasks in DeepMind Control Suite. The models are evaluated every 5k steps for 10 episodes, and the means and standard deviations are computed across 10 episodes across 4 seeds. The dashed lines are the asymptotic performances of 5M steps of Dreamer mentioned in (Hafner et al., 2019). The results suggest that NODA is very promising, in the sense that it can be easily combined with different MBRL methods, especially auto-encoder-type methods, by simply modifying their transition model.

