# OpenReview forum: "Model-based Reinforcement Learning with a Hamiltonian Canonical ODE Network"
_ICLR.cc/2022/Conference — ICLR 2022 Submitted_

### Official Review · Reviewer_grUX · 2021-11-02

**Correctness:** 2
**Technical Novelty And Significance:** 1
**Empirical Novelty And Significance:** 2
**Recommendation:** 3
**Confidence:** 4

**Main Review:**

My main concern is about technical soundness.

In section 3, the authors mentions
>For many RL environments such as MuJoCo (Todorovet al., 2012), the discretization of the continuous evolution is just the transition function.

This is incorrect. For tasks such as pendulum, cart-pole and double pendulum, the underlying dynamics is continuous and the discretization of that continuous dynamics is indeed the transition function. However, for many of the tasks that are tested in this paper (Ant-v3, HalfCheetah-v3, Hopper-v3, Walker), the underlying state evolution is not continuous, since these tasks involves collisions and contact, which results in instantaneous velocity change. In fact, the major contribution of Mujoco is its contact model. The authors seems to know Neural Event Functions (Chen et al, 2021), which specifically addresses this kind of discontinuities in the system. However, only bouncing balls examples are shown in Chen et al. 2021, as addressing contacts in deep learning for those systems in MuJoCo is difficult.

I'm very surprised that the authors didn't comment on this gap between their motivation/models and the systems in the experiments. The ODE-based model assumes continuous dynamics and this is an incorrect inductive bias for systems such as HalfCheetah, which involves contacts. The q and p learned in these systems have no physical meaning at all. Moreover, the authors couldn't justify their choice of Hamiltonian mechanics. A well-known property of Hamiltonian mechanics is that the energy conserves along a trajectory (assuming no action/control is applied). For Ant-v3, HalfCheetah-v3, Hopper-v3, Walker, energy is obviously not conserved due to the inelastic collisions.

The whole section 4 is dedicated to theoretical analysis. These analysis seems valid to me. However, they are not applicable to most systems in the experiments, since state evolution in those systems are not Lipschitz continuous (again because of contacts). From this perspective, I don't see how the theoretical analysis help us gain any insight into the application. A majority of the theory are extended from Asadi et al, 2018. However, Asadi et al, 2018 uses pendulum and cartpole as experiments to demonstrate how the theory works. And both pendulum and cartpole are both continuous dynamics, which means the theory applies to them. The authors seems to not realize that the theory is not applicable to the systems the study and inappropriately use the theory to gain technical soundness.



**Summary Of The Paper:**

This paper proposes a model of neural ODE auto-encoder, NODA, which incorporates Hamiltonian mechanics to learn a world model. The paper shows theoretical results of transition errors and value errors. NODA is tested on a range of RL tasks.

**Summary Of The Review:**

Despite the technical soundness, the proposed model seems to outperform existing models in the experiments. However, due to the concerns on technical soundness, I think this paper is not ready for ICLR yet. Thus I recommend rejection.

---

### Official Review · Reviewer_cybw · 2021-11-03

**Correctness:** 2
**Technical Novelty And Significance:** 2
**Empirical Novelty And Significance:** 2
**Recommendation:** 3
**Confidence:** 3

**Main Review:**

The idea of modeling the Hamiltonian in a latent is appealing and the best way of doing this is still largely unsolved in the community. This paper suggests a way to do this that augments existing RL methods (Fig 3/Table 1). My main concern with the paper is that it's difficult to interpret where the main improvement is coming from. The method (Alg 1) uses the Hamiltonian to augment data that an otherwise model-free agent is trained on, but the paper omits significantly related works that report similar results, such as [MBPO](https://arxiv.org/abs/1906.08253). Even without this, the improved sample efficiency this method makes at the beginning of training is marginal (the first 200k timesteps in Table 1/Figure 4), and no results are reported to the asymptotic performance. I think this is an important comparison to make, as if the Hamiltonian is truly modeling the system better than the existing dynamics models (e.g., Dreamer), then I would expect the asymptotic performance to be better as well.

Section 4.1 also talks about Lipschitz models and continuity. It's not clear to me that the MuJoCo environments are reasonably continuous, or are closed/conservative systems that can be modeled with a Hamiltonian, as there is contact and friction between the agents and the floor.

For one last minor comment: Definition 3 cites Arjovsky et al. (2017) for the definition of the Wasserstein metric, but this metric dates back much further in the optimal transport community. Villani is a more standard reference.

**Summary Of The Paper:**

This paper considers Hamiltonian models for reinforcement learning, and suggests to use it in Alg 1 to augment the data a policy is trained on. Fig 2 shows that this approach can converge faster than Hamiltonian Neural Networks on pendulum tasks, and Fig 3/Table 1 compares to standard methods on data-limited MuJoCo tasks.

**Summary Of The Review:**

Modeling the Hamiltonian is a promising direction but I do not feel a proper comparison to related approaches has been made here, such as to MBPO or to Dreamer's asymptotic performance.

---

### Official Review · Reviewer_9nPp · 2021-11-05

**Correctness:** 2
**Technical Novelty And Significance:** 1
**Empirical Novelty And Significance:** 1
**Recommendation:** 3
**Confidence:** 5

**Main Review:**

If the paper does what it claims to do, it would be amazing. In this case, the authors solved the two major problems of physics-inspired deep networks (e.g., HNN & DeLaN), which are the assumption of observing generalized coordinates & forces as well as not including contacts. However, the authors do not provide any experimental validation of this claim and even the architecture shows that the proposed model architecture should not be able to model the locomotion tasks.

For example, consider the network architecture in Figure 1. This model cannot describe the walker or hopper dynamics as these dynamics include contacts and are dissipative (assuming a(t) = \forall t). However, the architecture in Figure 1 is conservative, which means that the walker and hopper would need to bounce ground plane (which makes no sense). Therefore, this model architecture is not suitable for the performed tasks and should not work.

The theoretical evaluation in Section 4 has no relevance. Unfortunately including random theorems, which don't provide any insights, has become an annoying habit in too many ML papers. Could the authors please elaborate on why these theorems are useful or have any relevance? Furthermore, I even question the underlying assumption of Lipschitz dynamics as the dynamics model is applied to locomotion tasks that have discrete switching points at contact. If the theorems have no relevance these should be removed from the paper. It would be beneficial if the paper discussed the benefits and disadvantages of the model instead.

The experimental section also does not evaluate the proposed model architecture, it just wraps everything together in the model-based RL paradigm, which evaluates the proposed dynamics model architecture impossible. Therefore, one cannot make any claims about the dynamics models when comparing the learning curves. The advantage of the NODA model is also not significant. One would need to perform different experiments to isolate the impact of the model, which goes beyond comparing learning curves as these include many different aspects. The authors would need to provide a quantitative and especially QUALITATIVE evaluation of the learned dynamics model. Therefore, the authors need to provide video renderings of the model predictions. Furthermore, ablation studies are needed that look at when these models will break? How many steps can they predict?

Minor details:
- Eq 2 should be one line
- The description of the Noda + MFRL (which btw. is not MFRL afterwards) is a bit unclear. From the main paper, I cannot understand how the imagined data is used? Is the policy only trained on imagined data like dreamer or more in the direction of MBPO where true and imagined data is mixed.
- what is the real pendulum? Where does the data come from?
- The submitted code does not contain the data, e.g., learned models & datasets


**Summary Of The Paper:**

The paper proposes a variant of a Hamiltonian neural network for learning dynamics models for model-based RL. The proposed network architecture uses encoders/decoders to map the system state x = [q, p] from the observations s and vice versa. In the latent state, a standard HNN is used to compute dx/dt to increment the time step.

In the experiments, the paper applies this dynamics model to model-based rl and wants to show that this specific dynamics model increases the sample efficiency and leads to greater performance. It is especially noteworthy that the domains include locomotion tasks that include many contacts.

**Summary Of The Review:**

In summary, the paper should be rejected as the contribution on physics-inspired networks is marginal and the claims of the paper are not backed-up by experiments. Just adding a model and comparing the learning curves is not sufficient to isolate the impact of the proposed model learning approach.

---

### Decision · Program_Chairs · 2022-01-20

**Decision:**

Reject

**Comment:**

The reviewers raised concerns and the authors have not provided a response. All reviewers concur that this paper should be rejected at this time, and I agree.